# Analysis of Retractions in Nursing from Publications Between 2000 and 2024: A Bibliometric Analysis Using Retraction Watch

**DOI:** 10.3390/nursrep15100349

**Published:** 2025-09-26

**Authors:** María Paz Contreras-Muñoz, Cristian Zahn-Muñoz, Elizabeth Solís-Albanese, Ezequiel Martínez-Rojas

**Affiliations:** 1Departamento de Salud, Universidad de Los Lagos, Osorno 5290000, Chile; mariapaz.contreras@ulagos.cl; 2Centro de Estudios Universitarios, Universidad de Los Lagos, Osorno 5290000, Chile; cristian.zahn@ulagos.cl; 3Facultad de Ciencias de la Salud, Universidad Arturo Prat, Iquique 1100000, Chile; elsolis@unap.cl; 4Vicerrectoría de Investigación e Innovación, Universidad Arturo Prat, Iquique 1100000, Chile

**Keywords:** nursing, retraction, Retraction Watch, scientific publication, scientific integrity

## Abstract

There has been a significant increase in scientific publications in recent years, and the nursing field has been no exception. Consequently, the number of publications containing errors that lead to document retractions has also increased. It is essential to understand and delve into this phenomenon within nursing research. **Objective:** This study aims to identify and analyze the retractions of scientific publications in nursing research worldwide between 2000 and 2024. **Methodology:** This is a descriptive and cross-sectional study with a bibliometric approach. Data were collected using the Retraction Watch database, from which 408 retracted documents related to nursing research were extracted. **Results:** Over the last 25 years (2000–2024), a total of 408 documents in the nursing field have been retracted, with the majority concentrated in the 2020–2024 period, accounting for 84.8%. Ethical misconduct was the cause of retraction in 87.3% of the cases. Of the 408 retracted documents, 42.6% involved human participants in research or control groups, totaling 21,369 patients who were part of flawed studies. **Conclusions:** It is crucial that nursing research remains rigorous and adheres to bioethical standards, as these guide evidence-based nursing practice. Flawed literature can have significant consequences for patient health and care.

## 1. Introduction

In recent decades, the nursing field has experienced a notable increase in scientific output [1]. This advancement aims to address the concern raised by Gómez [2], who argues that, in the absence of sufficient discipline-specific evidence, nursing decisions are often based on knowledge produced by other professions. Consequently, nursing faces the ongoing challenge of developing robust scientific knowledge [3], reinforcing professional autonomy, and providing a sound foundation for clinical decision-making [4].

The challenges inherent to scientific production have contributed to establishing nursing as a robust and continuously evolving discipline [5,6]. This advancement has been driven mainly by the sustained growth in research output, which has reinforced the theoretical foundation of the field. In turn, this development has supported the creation of models and theories capable of describing, predicting, prescribing, and controlling phenomena specific to the discipline [7]. The scientific strength of nursing lies in the integration of practical expertise with an interdisciplinary perspective [8], where knowledge emerges from the dynamic interplay between theory and practice [9]. In this context, professionals who base their decisions on scientific evidence act responsibly, strengthen their professional identity, and foster excellence through the ongoing development of knowledge [4].

Scientific research requires the application of rigorous methodologies, encompassing everything from the formulation of hypotheses and the execution of studies to the interpretation of results [10]. In nursing, as in other health science disciplines, research must be conducted by ethical principles and moral values intrinsic to the profession [11]. Applying bioethics is fundamental to ensuring good practices and maintaining scientific integrity [12], which is a key requirement for preserving the credibility of the scientific system and represents an unavoidable responsibility for researchers [13]. Nevertheless, there are still cases where these standards are not upheld, whether due to unintentional errors or deliberate misconduct by those conducting the research.

As in other scientific disciplines, nursing is not exempt from producing flawed literature, as demonstrated by recent studies [14,15]. Such publications can have serious consequences: they compromise public health [16], pose a threat to the credibility and integrity of the scientific system [17,18], hinder the development of future research based on inaccurate results or conclusions [18], and endanger clinical practice [19] by increasing the likelihood of implementing inappropriate therapies [20]. In nursing, these impacts may lead to a decline in the quality of care, both in the prevention and treatment of health conditions—core research objectives in the discipline [21]. To mitigate these risks, the scientific community relies on corrective mechanisms—such as the retraction of publications—that alert researchers to the invalidity of specific studies [22].

Scientific retractions have become essential for addressing this issue [22]. They represent a formal process by which a journal or publisher withdraws a previously published article due to serious errors in the research or publication process [23]. This corrective measure enables the removal of fraudulent studies or those containing significant flaws that compromise the integrity of the scientific record [24]. Retractions play a critical role in safeguarding academic integrity by alerting readers to unreliable findings—whether stemming from misconduct, plagiarism, ethical violations, or methodological errors [25,26].

The retraction of scientific publications is a growing phenomenon [27,28] that has generated increasing interest within the academic community in better understanding its causes, dynamics, patterns, and consequences [22,29]. This rise is multifactorial and may be linked to several factors: the growing volume of scientific output, the expansion in the number of active journals [17] and those that actively issue retractions [27], and the availability of more advanced tools for detecting scientific misconduct [20,30]. Additional contributing factors include increased scrutiny of published research [31] and the lowering of publication barriers, both of which may influence the overall quality of the editorial process [32].

Furthermore, there is growing concern that the new scientific publishing culture is exacerbating the flawed research problem [33]. In this new context, publications are no longer exclusively oriented towards the generation and dissemination of knowledge but also function as tools to evaluate the professional performance of researchers [34], access additional rewards, gain recognition within the scientific community, achieve academic success, or improve personal image [35]. As a consequence of this cultural transformation, associated phenomena have emerged that generate adverse effects, such as the pressure to “publish or perish” [36,37], the proliferation of predatory journals [38,39], and the growth of so-called scientific article mills [40,41]. This pressure has led to increased academic productivity, but not necessarily to improved credibility or quality of publications [42,43]. While professional recognition and incentives are legitimate [11], they should not be achieved at the expense of academic integrity.

Although numerous studies have shown that most scientific retractions are due to misconduct, a significant portion also result from unintentional errors [44]. For instance, Hasselmann et al. [22] reported that 64% of retracted articles were attributed to ethical misconduct, with plagiarism being the most common reason. Similarly, Moylan and Kowalczuk [45], in their analysis of retractions in BioMed Central journals, found that 76% were due to misconduct, particularly fraudulent peer review and plagiarism. Campos-Varela et al. [46] observed that 65.3% of retractions in biomedical journals were linked to misconduct, with a higher prevalence in low-impact journals (73%) compared to high-impact journals (61%). Likewise, Campos-Varela and Ruano-Raviña [17] noted that 65.3% of retractions indexed in PubMed between 2013 and 2016 were related to scientific misconduct.

In contrast, fewer studies have identified more retractions caused by unintentional errors. For example, Nath et al. [18], in their analysis of retracted clinical and basic science articles indexed in Medline between 1982 and 2002, found that errors accounted for 61.8% of retractions, while misconduct accounted for only 27.1%. Wray and Andersen [33] reported that 51% of retractions resulted from errors, 35% from misconduct, and 14% were ambiguous cases.

In recent years, there has been a growing interest in understanding the reasons and patterns behind scientific retractions, particularly in disciplines within the health sciences. Studies have been conducted in various areas such as emergency medicine [47], obstetrics [19], surgery [31,48], cancer and oncology [49,50], genetics [51], radiology [52], orthopedics [53], neurology [54], and endocrinology [55]. However, the number of available studies in the specific nursing field remains limited and fragmented [56]. Analyzing retractions in the health sciences is particularly relevant, as it enables evaluating the quality of scientific practices across disciplines, thereby helping to strengthen the integrity of academic output and prevent future misconduct.

This research aims to analyze the retractions of scientific publications in the field of nursing at a global level, considering their temporal evolution, the main patterns and underlying reasons, the countries of affiliation of the authors involved, and the time elapsed between publication and retraction. This analysis seeks to identify trends, editorial practices, and recurring behaviors that offer a deeper understanding of the dynamics underlying the production of flawed literature in the discipline. Additionally, the study aims to shed light on the impact of retractions on the credibility of scientific knowledge and evidence-based clinical practice. In this context, the research aspires to provide valuable insights that strengthen the culture of scientific integrity in nursing and contribute to developing preventive, educational, and regulatory strategies to improve research quality.

## 2. Materials and Methods

### 2.1. Study Design and Data Sources

This study is descriptive and cross-sectional, with a bibliometric approach. Its purpose is to analyze retractions of scientific publications in the field of nursing published between 2000 and 2024.

Data were collected using the Retraction Watch (RW) database, downloaded from Crossref. This source was selected because it is publicly accessible [57] and is recognized as one of the most comprehensive databases with the most significant coverage in scientific retractions [58,59].

In addition, information was collected from other sources. To determine the journal quartile corresponding to each document’s year of publication, the Scimago Journal Rank (SJR) index was used, and the appropriate quartile was assigned based on the reference year. Citation counts per document were retrieved from Google Scholar, given its broader coverage compared to other scientific databases [60]. To estimate the total scientific output in the field of nursing between 2000 and 2024, the Scopus database was consulted using the following search command:

SUBJAREA (nurs) AND PUBYEAR > 1999 AND PUBYEAR < 2025 AND (EXCLUDE (DOCTYPE, “er”) OR EXCLUDE (DOCTYPE, “tb”))

The following search query was used to determine the number of publications per country in the nursing discipline:

SUBJAREA (nurs) AND PUBYEAR > 1999 AND PUBYEAR < 2025 AND AFFILCOUNTRY (“United Kingdom”) AND (EXCLUDE (DOCTYPE, “er”) OR EXCLUDE (DOCTYPE, “tb”))

### 2.2. Sample Selection

The Retraction Watch (RW) database was downloaded on 15 October 2024, at which point it contained 52,900 retracted publications. A specific filter was then applied to identify retractions related to the health sciences, yielding 16,210 records (30.6%). Subsequently, the dataset was filtered by discipline, selecting only the nursing category, which resulted in 411 retracted publications. All retractions published before the year 2000 were excluded from this set, leading to a final sample of 408 retractions included in the present analysis (Figure 1).

### 2.3. Data Collection

The information collected for each retraction included the title of the publication, document type (original article, literature review, meta-analysis, book chapter, clinical case, case report, or conference paper), number of authors, corresponding author’s institutional affiliation, country of the institution, journal name, publisher, publication date, retraction date, and reason for retraction, based on the classification established by Retraction Watch [61].

In addition, data were recorded, such as the journal’s quartile at the time of publication (according to Scimago Journal Rank) and the number of citations received before and after the retraction. Pre-retraction citations were counted retrospectively up to the year the retraction was issued, while post-retraction citations included those registered from the year following the retraction. Through individual document review, studies involving human participants were identified, and the number of participants was also recorded; this information was entered into dedicated columns within the database. Lastly, the number of days, months, and years elapsed between each document’s publication and retraction was calculated and systematized in separate fields.

### 2.4. Systematization and Categorization of Retraction Reasons

Regarding the reasons for retraction, a data-cleaning process was conducted to exclude causes unrelated to errors attributable to the article (e.g., third-party investigations, editorial inquiries, and withdrawn and replaced articles, among others). Specifically, retractions labeled “withdrawn” were mainly associated with systematic reviews removed due to a lack of author interest or the inability to update the content. In response, a new retraction category was introduced: “Outdated Review”.

Subsequently, a categorization process was carried out, grouping the retraction reasons into three significant categories, following classifications used in previous studies [18,32,33]:Misconduct: Includes cases of plagiarism, article duplication, lack of informed consent, fake peer review, random content generation, falsification, fabrication, and manipulation of images, data, and/or results, among others.Unintentional error: Includes errors in methods, analysis, text, conclusions, and/or results, among others.Other reasons: Includes journal or editor errors, outdated reviews, and copyright-related issues.

The retraction notices were individually read to assign each retraction to a category, given that some reasons may be interpreted as unintentional errors or as misconduct, depending on the context. Two authors independently assigned each retraction to one of the three categories, achieving agreement in 383 cases (93.9%). Discrepancies were resolved through discussion and consensus.

As previously noted, some reasons for retraction are inherently ambiguous. For example, the reason for “unreproducible results” could stem from a methodological flaw (i.e., unintentional error) or a serious ethical violation (i.e., misconduct). If this reason appeared alongside “analysis error”, it was categorized as unintentional since the irreproducibility resulted from prior technical mistakes. Conversely, if it appeared with “data manipulation”, the case was classified as misconduct, as the irreproducibility reflected deliberate scientific fraud.

### 2.5. Statistical Analysis

The data were organized and systematized in a Microsoft Excel database, allowing for their cleaning, coding, and preparation for subsequent analysis. A descriptive analysis was conducted using IBM SPSS Statistics software, version 21, to characterize the retracted publications. In addition, the nonparametric Mann–Kendall time trend test was applied using Python 3.10 software. This test was used to assess whether there was a significant increase in the number of nursing retractions between 2000 and 2024. The relevance of its use lies in the marked increase recorded in 2023.

## 3. Results

### 3.1. Evolution of Retractions

The Retraction Watch (RW) database records 52,990 retracted scientific publications, of which 16,210 (30.6%) pertain to disciplines related to the health sciences. Within this group, 2.5% of the retractions are specifically associated with the field of nursing (40).

Over the past 25 years (2000–2024), 408 documents in the nursing field have been retracted. A notable concentration of retractions was observed in 2020–2024, during which 346 documents were withdrawn, representing 84.8% of the total (Figure 2).

The trend peaked in 2023, with 271 retractions in a single year. This spike was primarily driven by a mass retraction initiative led by the Hindawi publishing house between 2022 and 2023, during which over 8,000 publications were withdrawn—254 of them in the nursing field.

The influence of Hindawi is particularly evident in recent years: in 2022, 60.7% of nursing-related retractions were linked to this publisher; in 2023, the proportion rose to 94%; and in 2024, it stood at 70%.

Despite the increase in retracted publications within the nursing field, these represent a tiny proportion of the total scientific output. According to data from the Scopus database, 1,198,447 nursing-related publications were recorded between 2000 and 2024, excluding documents categorized as “retracted” or “erratum” and those published in 2025. Based on these figures, an average of 3.4 articles are estimated to be retracted per 10,000 publications, indicating that although the phenomenon exists, its relative magnitude remains limited.

The nonparametric Mann–Kendall test was applied to assess whether there was a significant increase in retractions during the analyzed period. The result showed a *p*-value of 0.000, which allowed us to reject the null hypothesis of no trend and assume sufficient evidence to support a significant increase.

### 3.2. Features of the Documents and Sources of Publication

Regarding the type of document, 69.9% of retractions in the nursing field were linked to original research articles, followed by clinical studies at 20.6% and literature reviews at 5.1% (Table 1).

Regarding the quality of the publication sources, of the 408 documents analyzed, 257 (63%) were published in Q2 journals, followed by 15% in Q1, 13.7% in Q3, and 1.5% in Q4 journals. Additionally, it is worth noting that eight articles—representing 2% of the total sample—were published in journals not indexed in the Scimago Journal Rank (SJR).

Table 2 lists the five journals with the highest number of identified retractions. Sixty-nine publication sources were associated with the 408 retracted documents, with these five journals accounting for 56.7% of the retractions.

Notably, the journals *Evidence-Based Complementary and Alternative Medicine*, *Computational and Mathematical Methods in Medicine*, and *Journal of Healthcare Engineering* each had more than 60 retracted publications.

It is important to highlight that three of these five journals have been discontinued from the Scimago Journal Rank (SJR), reflecting potential issues with editorial quality. Furthermore, none of the five journals with the highest incidence of retracted articles are specific to nursing.

### 3.3. Reasons for Retractions

An analysis of scientific retractions in nursing reveals a concerning reality: 87.3% of the documents were retracted due to scientific misconduct. This category includes serious issues such as plagiarism, data falsification, and other behaviors that compromise research integrity.

On the other hand, only, only 6.1% of retractions were related to honest errors, meaning unintentional mistakes that, while affecting the credibility of the results, did not stem from ethical violations. To a lesser extent, 3.7% of cases were attributed to other reasons, such as editorial errors or administrative issues.

However, 12 documents remain for which the reason for retraction was not disclosed, leaving open questions about the factors that led to their withdrawal (Figure 3).

To illustrate the impact of Hindawi mass retractions in 2023, it is worth noting that of the 254 nursing-related articles retracted that year, 247 (97.2%) were withdrawn due to issues related to compromised peer review.

When retraction categories are analyzed, excluding the papers associated with the 2023 Hindawi mass retraction, the results in Figure 4 reveal a notably different pattern. Although misconduct remains the leading cause of retraction, its proportion drops significantly from 87.3% to 66.9%, reducing 20.4 percentage points. This shift suggests that the prevalence of Hindawi-related cases may have partially skewed the discipline’s overall distribution of retraction reasons.

This situation highlights the need to reflect on ethical practices in scientific publishing and strengthen control and review mechanisms. This will not only prevent unintentional errors but also eradicate unethical practices that undermine trust in nursing research.

Table 3 presents a list of the 15 most common reasons for retraction. The most notable are concerns or issues with peer review, reported in 295 documents (74.5%); concerns or issues related to data, present in 271 documents; unreliable results, mentioned in 269 documents; and concerns or issues with references or attributions, observed in 266 publications.

### 3.4. Authorships

The analysis of retracted documents identified 1552 authorship signatures, corresponding to 1434 distinct authors. The distribution of the number of authors per document reveals an interesting pattern: the most significant proportion of articles (41.9%) had two to three authors, followed by those with four to five authors (29.7%). On the other hand, single-author documents accounted for 9.6%, and 18.8% of publications had more than six authors.

One particularly noteworthy aspect is the recurrence of specific authors. Among the 1434 authors, 96 (6.7%) appeared repeatedly, meaning they were involved in more than one retraction. These authors collectively accounted for 214 authorship signatures, averaging 2.23 signatures per recurring author (Table 4).

Table 5 analyses the 10 countries with the highest number of first authorships affiliated with institutions from these nations. China accounts for an overwhelming 77.7% of first authorships, followed by the United Kingdom (3.2%) and the United States (2.2%), though with significantly lower proportions.

Authors from 30 different countries were identified as first authors. However, only four Latin American countries appeared in this category, highlighting the low regional representation of primary authorship in retracted documents.

Table 6 lists the top 10 countries with the most retracted documents based on author affiliation. In addition to the number of retracted documents, the table includes the total number of publications in nursing and the retraction percentage for each country.

China tops the list with 320 retractions, representing a rate of 58.5 retractions per 10,000 publications and a retraction percentage of 0.58%. This figure is significantly higher than any other country in the sample, suggesting an unusual concentration of retractions in that region.

The Philippines stands out for having one of the highest retraction rates (30.9 per 10,000 publications) despite having a low absolute number of retractions (5), which reflects a proportionally high incidence due to its limited research output (1616 publications).

The United States and the United Kingdom have the highest numbers of publications (373,642 and 122,377, respectively), yet their retraction rates are low (0.3 and 1.2 per 10,000 publications).

### 3.5. Citation of Retracted Documents

Of the 408 analyzed documents, 388 (95.1%) have been cited, accumulating 5693 citations (Table 7), corresponding to an average of 13.95 citations per document. A closer look at these citations reveals an interesting pattern: 3,005 citations (52.8%) occurred before or during the year of retraction, while the remaining 47.2% took place at least one year after the retraction. This indicates that these documents circulated within the scientific literature even after being withdrawn.

A total of 60.3% of the cited articles (234 documents) received citations after being retracted, raising concerns about the continued dissemination of flawed or invalid scientific knowledge. An analysis of citation distribution reveals a distinct pattern: the majority of these documents received a relatively low number of post-retraction citations, with 63.9% accumulating between one and five citations. However, a subset of retracted articles had a more substantial impact: 16.8% were cited between 6 and 10 times, 8% received between 11 and 20 citations, 4.4% were cited between 21 and 49 times, and 3.6% accumulated between 50 and 99 citations. Notably, 13 articles (3.4%) were cited more than 100 times, positioning them as highly influential pieces despite their retracted status.

### 3.6. Impact on Patients

More than half of the 408 analyzed documents and 234 studies (57.4%) did not include patients in their research, indicating a predominance of studies without direct human participation. However, in the remaining 42.6%, human subjects were involved in the research.

Within this group, interesting patterns emerged regarding the magnitude of the samples: in 82 documents (47.1%), participation involved fewer than 100 patients, while in 72 studies (41.4%), the number of participants ranged between 100 and 199. Finally, a smaller group of 20 studies reached higher figures, with more than 200 patients (Figure 5).

The study “Data Analysis of Nursing Effects in Pediatric Gastroenterology Department under High Content Image Analysis Technology” stands out for its sample size and the ethical controversies that led to its retraction. With 2465 participants, all under the age of 3 years, the study divided the children into two groups: an observation group, which received routine nursing care, and a control group, which was subjected to nest nursing and quality nursing.

However, serious ethical irregularities emerged behind these findings. The study was retracted due to the lack of informed consent and the absence of approval from the institutional ethics review committee, two fundamental bioethical principles that ensure the dignity and safety of participants.

This case was not an isolated incident. In analyzing the 174 studies involving human patients, bioethical violations were concerning. In 69 studies (39.7%), the institutional ethics committee did not approve the studies, while the lack of informed consent was the reason for retraction in 38 cases (21.8%). Additionally, in 31 documents (17.8%), retractions were linked to concerns regarding the well-being and integrity of human subjects.

### 3.7. Retraction Time

In the analysis of the 408 retracted documents, it was identified that four studies lacked information on the exact date of their retraction. The remaining cases revealed interesting patterns regarding the time between publication and retraction. A total of 138 documents were retracted less than a year after publication, while 214 studies had a retraction time of between 1 and less than 2 years. Additionally, 31 documents took between 2 and less than 5 years to be withdrawn, and 21 studies were retracted more than 5 years after publication (Table 8).

Exceptionally, seven documents took more than 10 years to be retracted. The most extreme case corresponds to a study published in 1998 and retracted in 2014, with an interval of 15.79 years, making it the document with the longest recorded retraction time.

The overall average retraction time was 1.66 years, but notable differences were identified when broken down by retraction cause. Documents withdrawn due to misconduct had an average retraction time of 1.42 years, heavily influenced by the massive retractions of 2023. During this period, of the 271 retracted documents, 222 (81.9%) had been published just a year earlier, in 2022.

On the other hand, documents retracted due to honest errors had an average retraction time of 2.33 years, a more extended period compared to misconduct cases. Finally, studies retracted for other reasons exhibited a significantly higher average of 5.17 years between publication and retraction.

## 4. Discussion

The objective of this study was to analyze retractions in nursing, observing their evolution over time and describing their main characteristics. Globally, retractions in this discipline remain relatively low, accounting for only 2.5% of all health-related retractions. However, this number should not be underestimated, as recent years have seen a significant increase, particularly driven by events such as the mass retraction by Hindawi in 2023, where 254 documents were withdrawn due to scientific misconduct.

The phenomenon of mass retractions initiated by Hindawi in 2023 represents the largest corrective action in the history of scientific publishing undertaken by an academic publisher [44]. This unprecedented process was triggered by the detection of systematic patterns of editorial manipulation, particularly within special issues, prompting the publisher to temporarily halt these publications and implement a rigorous evaluation protocol [62].

Hindawi’s experience should be seen as a warning sign of the potentially vast volume of flawed literature that remains unretracted. The identification of manipulation patterns underscores the presence of increasingly sophisticated fraudulent schemes capable of bypassing traditional editorial safeguards [63]. In this context, Hindawi’s response is a corrective measure and an urgent call to self-examine for other scientific publishers.

Our findings reveal that most retracted documents correspond to research articles (69.6%) and clinical studies (20.6%). These publications were distributed across 69 different journals, 98% of which were indexed in the Scimago Journal Rank (SJR) at the time of publication. Notably, the highest rate of retractions originated from journals classified in the Q1 and Q2 quartiles (78%). This result contrasts with previous studies, such as Al-Ghareeb et al. [14], which suggested a higher likelihood of retractions in lower-impact journals. These results suggest that although article retractions can occur across all levels of editorial quality, they appear to be more prevalent in high-impact journals—possibly due to the greater visibility and more rigorous scrutiny these publications undergo [63].

A total of 87% of the retractions identified in this study were attributed to scientific misconduct, a figure strongly influenced by the large-scale vetting process conducted by Hindawi in 2023. Specific causes included paper mills, peer review manipulation, randomly generated content, and the absence of informed consent. Particular attention should be given to fraudulent peer review practices and the rise of paper mills, which have significantly increased in recent years [64,65]. When the mass retractions by Hindawi in 2023 are excluded from the analysis, misconduct still accounts for 66.9% of retractions—a finding consistent with previous studies reporting similar proportions: 68% [56] and 69% [15].

This situation is alarming, especially considering that professionals in health sciences, particularly nursing, rely on academic literature as evidence for clinical decision-making. Inaccurate findings compromise clinical practice and put patient safety at risk. Moreover, the findings reveal an ongoing citation of retracted documents, contributing to the continued dissemination of flawed knowledge within the scientific literature. This issue not only undermines the credibility of science but may have direct consequences for advancing knowledge and professional practice—particularly in a field as sensitive and impactful as nursing.

When analyzing the country affiliation of first authorships, China led with 77% of cases, which aligns with Al-Ghareeb et al. [14], who reported that 31% of corresponding authors in retracted nursing and obstetrics papers were affiliated with Chinese institutions. These data align with other health sciences studies, highlighting China’s significant challenges concerning the integrity of its scientific publications [44,66,67].

Regarding retraction time, we found interesting results. While previous studies suggested that misconduct-related retractions tend to take longer, in our research, the “other reasons” category had the longest retraction times, primarily influenced by outdated Cochrane reviews, which may be withdrawn after 5.5 years of inactivity [68]. The overall average retraction time was 1.66 years, a figure lower than that reported by Al-Ghareeb et al. but higher than the 12-month average found by Joaquim et al. [15] and very similar to the 1.89 years reported by Nicoll et al. [56].

Our data show that documents retracted after over 5 years accumulated an average of 57 post-retraction citations. In contrast, those withdrawn between 2 and 5 years had 13 citations, and those retracted within less than 2 years received an average of 3.2 citations. Therefore, the time between the publication of an article and its eventual retraction is a crucial factor that influences the extent to which errors or misinformation can spread within the scientific literature. As the data indicate, articles that remain unretracted for extended periods tend to accumulate a significantly higher number of post-retraction citations. This is particularly concerning, as it facilitates the continued circulation of invalid conclusions, fabricated data, or flawed methodologies. Such a pattern underscores the urgent need to enhance retraction communication mechanisms and to promote a culture of continuous, critical, and timely engagement with the scientific literature to safeguard the integrity of academic knowledge.

Finally, to assess patient impact, we conducted an analysis similar to Steen [69], who found that 9189 patients participated in retracted primary studies, and 70,501 were involved in secondary studies. In our study on nursing retractions, we identified 21,369 patients who participated in 164 retracted studies, with an average of 130 patients per document. This situation raises significant ethical concerns about the integrity of research processes and the accountability of institutions. Particularly troubling is the potential ripple effect when findings from retracted studies are used to inform clinical practice, thereby putting patient safety and well-being at risk. This scenario highlights the urgent need to reinforce scientific validation mechanisms and enhance the critical appraisal skills of health professionals, ensuring that clinical research is conducted and applied by the highest ethical and methodological standards.

## 5. Limitations

The first limitation concerns retraction timing, as the average delay for retracting a nursing publication is approximately 19.8 months. As a result, the number of retractions corresponding to 2023 and 2024 is likely not yet fully captured in the available data. This may lead to an underestimation of recent retractions and affect the interpretation of observed temporal trends.

A second important limitation is the lack of standardization in retraction notices regarding the reasons for withdrawal. In some cases, the notice is clear and provides sufficient detail to understand the rationale behind the retraction. However, in other instances, the information is ambiguous or incomplete, making it challenging to determine key elements such as the specific cause or the origin of the retraction—whether institutional, editorial, or author-related. Additionally, there are cases where no formal notice is available, which prevents a definitive understanding of the reasons behind the retraction.

## 6. Conclusions

Our study shows that retractions of nursing publications represent a smaller percentage of total publications in health sciences (2.5%). Until 2021, they remained below 10 retractions per year; however, the increase starting in 2022 is significant. Due to this increase, it is imperative that knowledge production in this field strictly adhere to bioethical principles and maintain the utmost methodological rigor, not only to ensure the validity of the results but also to prevent errors that could negatively affect professional practice and consequently harm patients.

The prevalence of retractions due to intentional scientific misconduct (87.3%), such as manipulation of peer review, random content generation, lack of informed consent, and use of “article mills,” calls on the scientific community to strengthen scientific integrity, train responsible researchers, and improve editorial oversight mechanisms. These actions represent unavoidable tasks to ensure that knowledge production in nursing grows in volume, veracity, and credibility.

Another finding that should be a cause for reflection is that, on average, erroneous literature in this area takes 1.66 years to retract, during which flawed publications negatively impact medical literature and practice. For example, as shown in our results, a high percentage of publications were cited, and 60.3% continued to be cited after they were retracted.

Finally, the case of mass retractions at Hindawi Publishing House demonstrates that the integrity of academic literature cannot be maintained solely through reactive measures or isolated efforts. Publishers must adopt a proactive approach by systematically auditing their archives, particularly those segments most vulnerable to manipulation, such as special issues and externally managed peer-review processes. Only through continuous and active monitoring can trust in the scientific publishing system be restored.

## Figures and Tables

**Figure 1 nursrep-15-00349-f001:**
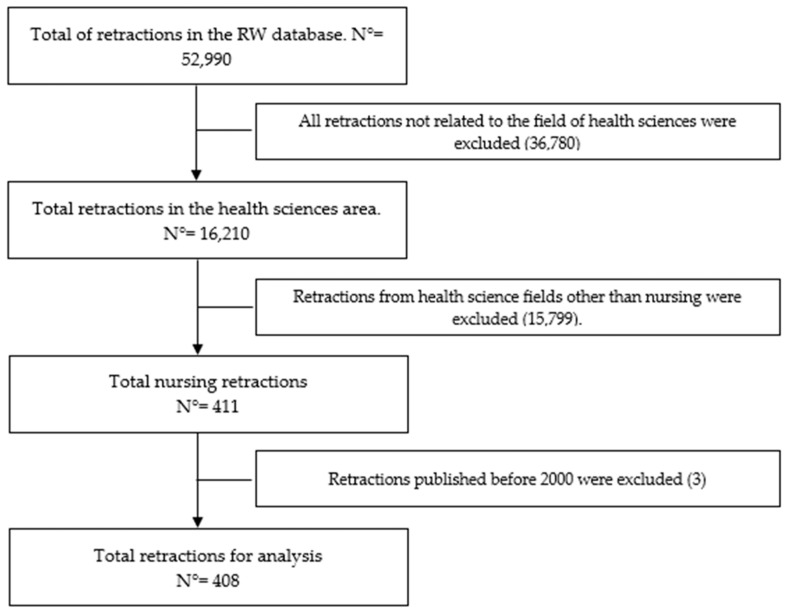
Flow diagram of the sample selection process.

**Figure 2 nursrep-15-00349-f002:**
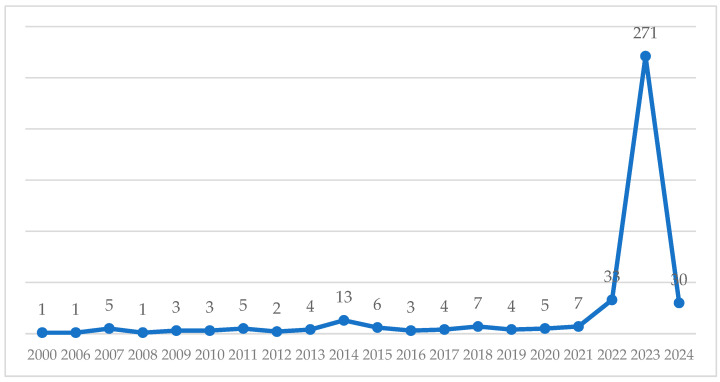
Evolution of retractions in nursing between 2000 and 2024.

**Figure 3 nursrep-15-00349-f003:**
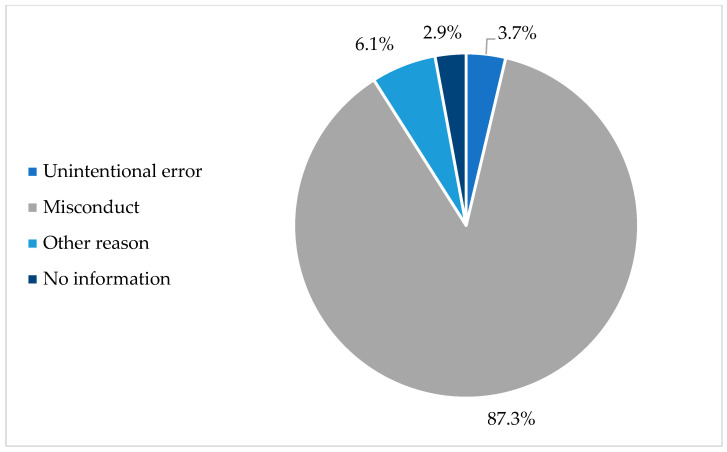
Retraction rates of documents by category.

**Figure 4 nursrep-15-00349-f004:**
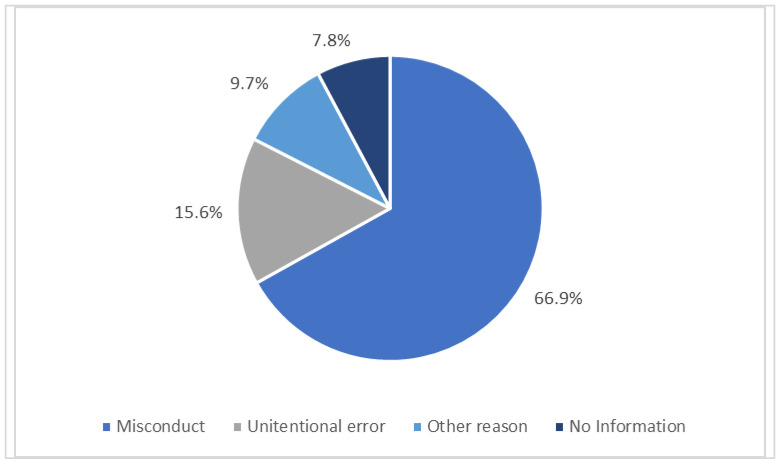
Retraction rates by category (excluding Hindawi 2023 retractions).

**Figure 5 nursrep-15-00349-f005:**
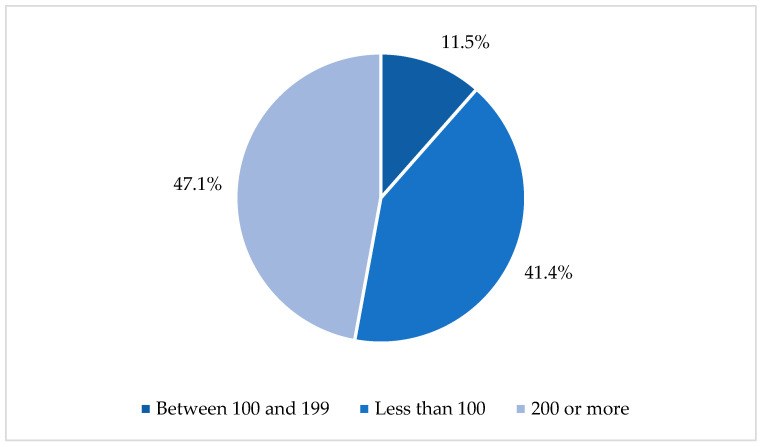
Number of patients by range.

**Table 1 nursrep-15-00349-t001:** Characteristics of documents and publication sources.

Category	No. of Documents	% of Documents
Type of Document		
Case Report	2	0.5%
Clinical Study	84	20.6%
Editorial	3	0.7%
Conference paper	5	1.2%
Meta-analysis	8	2.0%
Original article	284	69.6%
Literature Review	22	5.4%
Journal Quality		
Q1	61	15.0%
Q2	257	63.0%
Q3	56	13.7%
Q4	6	1.5%
No quartile assigned	20	4.9%
Not indexed in SJR	8	2.0%

**Table 2 nursrep-15-00349-t002:** Journals with the highest number of retracted documents.

Source	No. of Documents	% of Documents	Quartile 2023
*Evidence-Based Complementary and Alternative Medicine*	68	16.7%	Discontinued in 2023
*Computational and Mathematical Methods in Medicine*	65	15.9%	Discontinued in 2023
*Journal of Healthcare Engineering*	61	15.0%	Quartile 2
*Contrast Media & Molecular Imaging*	19	4.7%	Discontinued in 2023
*BioMed Research International*	18	4.4%	Quartile 2

**Table 3 nursrep-15-00349-t003:** Most frequent reasons for retraction.

Reason	No. of Documents	% of Documents
Concerns/issues with peer review	295	74.5%
Concerns/issues with data	271	68.4%
Unreliable results	269	67.9%
Concerns/issues with references/attributions	266	67.2%
Concerns/issues with results	152	38.4%
Randomly generated content	140	35.4%
Lack of IRB/IACUC approval	133	33.6%
Paper mill	104	26.3%
Informed consent/patient consent: none/withdrawn	85	21.5%
Concerns/issues regarding the well-being of human subjects	46	11.6%
Duplication of article	15	3.8%
Error by journal/publisher	14	3.5%
Euphemisms for plagiarism	12	3.0%
Plagiarism of article	11	2.8%
Outdated review	10	2.5%

**Table 4 nursrep-15-00349-t004:** Characteristics of authorships in retracted documents.

Authors	No. of Authors	% of Total
1 author	39	9.6%
2 or 3 authors	171	41.9%
4 or 5 authors	121	29.7%
Between 6 and 9 authors	72	17.6%
10 or more authors	5	1.2%
**Author Recurrence**		
Non-recurrent authors	1.338	93.3%
Recurrent authors	96	6.7%
Total authors	1.434	100%

**Table 5 nursrep-15-00349-t005:** Ranking of the 10 countries with the highest presence of first authorships.

Country (First Author)	No. of Authors	% by Country
China	317	77.7%
United Kingdom	13	3.2%
United States	9	2.2%
Irán	9	2.2%
Turkey	7	1.7%
South Korea	6	1.5%
Australia	6	1.5%
Philippines	5	1.2%
Saudi Arabia	4	1.0%
Jordan	4	1.0%

**Table 6 nursrep-15-00349-t006:** Top 10 countries with the highest number of retractions based on author affiliation.

Country	No. of Retracted Documents	No. of Publications	Retraction Percentage	Retraction Rate per 10,000 Publications
China	320	54,741	0.58%	58.5
United Kingdom	15	122,377	0.01%	1.2
United States	12	373,642	0.00%	0.3
Iran	9	17,050	0.05%	5.3
Australia	7	62,035	0.01%	1.1
Saudi Arabia	7	6032	0.12%	11.6
South Korea	7	29,226	0.02%	2.4
Turkey	7	15,756	0.04%	4.4
Canada	5	50,564	0.01%	1.0
Philippines	5	1616	0.31%	30.9

**Table 7 nursrep-15-00349-t007:** Citations of retracted documents.

Category	Number of Citations	% of Citations
Cited Documents		
Cited articles	388	95.1%
Non-cited articles	20	4.9%
Number of Citations		
Citations before retraction	3005	52.8%
Citations after retraction	2688	47.2%
Total citations	5693	100%
Citation ranges		
1–5 citations	248	63.9%
6–10 citations	65	16.8%
11–20 citations	31	8.0%
21–49 citations	17	4.4%
50–99 citations	14	3.6%
100+ citations	13	3.4%

**Table 8 nursrep-15-00349-t008:** Retraction time of documents.

Range	No. of Documents	% of Documents
<1 year	138	33.8%
≥1 year and <2 years	214	52.5%
≥2 years and <5 years	31	7.6%
≥5 years	21	5.1%
No information	4	1.0%

## Data Availability

Data are available upon request from the corresponding author.

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
