# Peer review of "Analysis of Retractions in Nursing from Publications Between 2000 and 2024: A Bibliometric Analysis Using Retraction Watch"

_nursrep, 2025, doi:10.3390/nursrep15100349_

Round 1
Reviewer 1 Report
Comments and Suggestions for Authors
In this bibliographic and bibliometric report, retractions in worldwide nursing
research from 2000 to 2024 are investigated. A total of 408 retracted papers were retrieved from the RetractionWatch database, which was recovered from Crossref. The metadata of the journal, including publication date, authors, institutions, country, reasons for retraction, year of retraction, time from publication to retraction, and number of patients involved, where feasible, were collected. Categorization was applied according to the retraction reasons, such as misconduct, unintentional error, and other reasons unrelated to the authors’ responsibility (i.e., errors by the journal or editor). A sharp increase in retractions was observed after 2020 when nearly 85% of the total retractions occurred.
The authors suggested that this increase in retractions was the result of Hindawi
publishing house cleanup. Indeed, it appears that 62% of the retractions in nursing research were of publications in the Hindawi publishing house. The conclusion is that flawed evidence in nursing research directly compromises the quality of healthcare; therefore, alert measures of faulty reports, as well as ethical and methodological training of professionals, are necessary.
Concerns
1. This project is based on the RetractionWatch database, an excellent resource
for this purpose. However, it should be mentioned that these cases represent a
small fraction of the total publications in this field of research. According
to Scopus data, there are 1,208,533 documents in nursing research published
between 2000 and 2024. Thus, the 408 retractions represent a small fraction of
3 out of 10,000 papers.
2. Another interesting fact, according to the Scopus database, is that there
are 9,433 erratum cases in the corpus of 1,208,533 nursing research documents.
These cases illustrate the reflexes of healthcare investigators in responding
to errors.
3. The predominant problem with the retracted documents dataset analyzed is
the number of retractions from the Hindawi publishing house. This problem affects the whole RetractionWatch dataset. It is evident in Figure 1, with the sharp increase in retractions in 2023, when the Hindawi case arose, as well as in Table 2. Notably, all the top five journals with the highest number of retractions are Hindawi journals, and three of them were discontinued by 2023. Of course, this problem affects all scientific fields where Hindawi publishing was involved. The authors should consider examining the Hindawi cases separately from the rest of the retraction cases, as they may mask the evidence for them. It appears that 62% of the retractions in nursing research were of publications from the Hindawi publishing house, all occurring in 2023 due to the Wiley investigations into Hindawi's compromised peer reviews.
4. If the authors separate the Hindawi cases from the rest, there are 156 remaining retractions, a number in agreement with the Scopus dataset for the nursing research field. Scopus does not account for the batch effect of Hindawi papers since they were removed from the bibliography. This set of 156 retractions exhibits a normal distribution through time and different characteristics than the batch Hindawi retractions.
Author Response
Dear reviewer, Thank you very much for your feedback. The latest version of the article with all its corrections is attached. Best regards.

Reviewer 2 Report
Comments and Suggestions for Authors
Thank you for the opportunity to review this manuscript on an important subject that receives limited attention in the scientific literature. I have some remarks and suggestions for further improvement.
General:
- The focus of the study is on nursing science. However, the information in the introduction and discussion is not specific to nursing science but also to medical science and every other science in the field of sickness and health. Please be clear about what is specific to nursing science and discuss your results in light of different health science fields, such as medicine. Are papers by nurses more prone to be retracted than papers by medical doctors? And what does that mean? Or is that unknown? Such questions need to be addressed throughout the manuscript.
Introduction:
- The scope of the study is broad, focussing not only on misconduct but also on methodological flaws. The research question and methodology focus primarily on retracted studies. Please rewrite the introduction to where the specific subgroup of retracted studies is highlighted and where you elaborate more on specific features of nursing science in this context.
Method:
- Please provide more information about the selection and data-extraction process and which authors performed these steps (if there is more than one, please elaborate on the independence of this exercise).
- Please provide a more comprehensive definition of the categories of retracted studies.
Results:
- The presented results speak for themselves, but context is omitted here. For instance, most retractions are by authors affiliated with Chinese institutions, but how does this relate to the total proportion of studies in nursing science affiliated with Chinese institutions? This is important for interpretation and is necessary information for all reported results.
- Please elaborate on why Hindawi performed such a significant retraction project. This has a major influence on your findings and should receive appropriate attention.
Discussion:
- The mass retraction of a major publisher is highlighted as a reason for the significant increase. Did the pandemic have any influence on this matter? Also, comparison with other fields (on trend level) is necessary for interpretation.
- Please give information on the strengths and limitations of this study.
Author Response

(The authors gave the same response as above.)

Reviewer 3 Report
Comments and Suggestions for Authors
Comments and suggestions to the authors are detailed in the attached file.

Author Response

(The authors gave the same response as above.)

Round 2
Reviewer 1 Report
Comments and Suggestions for Authors
This bibliographic and bibliometric report investigates retractions in worldwide nursing research from 2000 to 2024. The authors recognized a substantial impact of the Hindawi publishing house retractions. These retractions were so massive that they affected the whole RetractionWatch dataset. The authors addressed the reviewers' concerns well.
Author Response
Dear Reviewer
The authors of this article would like to thank you for your time in reviewing the article submitted for review. In the second round, Reviewer 1 provided no comments suggesting modifications.
Reviewer 2 Report
Comments and Suggestions for Authors
Thank you for the opportunity to review this revised manuscript. Although a comprehensive letter to address all reviewers' remarks specifically, the manuscript improved substantially. I have no further comments and thank the authors for this interesting and important piece of work.
Reviewer 3 Report
Comments and Suggestions for Authors
The authors have addressed most of the recommendations made in the previous review, for which I congratulate them and thank them for their efforts.
However, there is still one relevant omission that affects the methodological quality of the study. In particular, I refer to the section on ‘2.5. Statistical Analysis' (line 225). Although this section has been incorporated with respect to the original version of the manuscript, it still does not clearly specify what type of statistical analysis has been applied to the study variables, nor the statistical tests used. Furthermore, the results presented do not explicitly reflect the analyses performed.
As an example, a time trend analysis (such as the Mann-Kendall test) could be included to examine the evolution of retractions over time, which would allow to identify the existence of a statistically significance trend in the annual frequency of such publications.
I am aware that the required modifications require additional effort on the part of the authors. However, I believe that they are essential to reinforce the methodological soundness of the study and to ensure that it meets the standards required in a rigorous bibliometric analysis, typical of an indexed scientific publication.
Author Response
We appreciate the detailed review of our manuscript, "Analysis of Retractions in Nursing Publications between 2000 and 2024: A Bibliometric Analysis with Retraction Watch."
We have carefully addressed the comments made by reviewer 2.
Their suggestions were incorporated using the nonparametric Mann-Kendall test. This incorporation resulted in modifications to a methodology section, the addition of a results paragraph, and improvements to the study's conclusion.
The modifications are highlighted in yellow. We hope we have satisfactorily addressed the editor's suggestion.